**Data Availability Statement:** dataset URL in a public repository: https://www.kaggle.com/vtsogli/unpredictability-what-where-when.

**Funding:** The author(s) received no specific funding for this work.

# Unpredictability of the "when" influences prediction error processing of the "what" and "where"

**Vera Tsogli** [1], **Sebastian Jentschke** [2], **Stefan Koelsch** [1,3] *

**1** Department of Biological and Medical Psychology, University of Bergen, Bergen, Norway, **2** Department of Psychosocial Science, University of Bergen, Bergen, Norway, **3** Max Planck Institute for Human Cognitive and Brain Sciences, Leipzig, Germany

* Stefan.Koelsch@uib.no

## Abstract

The capability to establish accurate predictions is an integral part of learning. Whether predictions about different dimensions of a stimulus interact with each other, and whether such an interaction affects learning, has remained elusive. We conducted a statistical learning study with EEG (electroencephalography), where a stream of consecutive sound triplets was presented with deviants that were either: (a) statistical, depending on the triplet ending probability, (b) physical, due to a change in sound location or (c) double deviants, i.e. a combination of the two. We manipulated the predictability of stimulus-onset by using random stimulus-onset asynchronies. Temporal unpredictability due to random onsets reduced the neurophysiological responses to statistical and location deviants, as indexed by the statistical mismatch negativity (sMMN) and the location MMN. Our results demonstrate that the predictability of one stimulus attribute influences the processing of prediction error signals of other stimulus attributes, and thus also learning of those attributes.

## Introduction

Learning is a fundamental property of nervous systems, and recent accounts from cognitive and computational neuroscience have established the underlying role of prediction for learning [1–5]. Within the framework of predictive coding (PC), learning is the generation of a predictive model of the world which is continually optimized by reducing prediction errors, i.e. any mismatches between incoming sensory signals and the predicted encoded input [6, 7]. Furthermore, in real life we tend to learn by forming predictions over several dimensions (e.g., location, intensity, timing). Thus, our predictive model encompasses a multitude of features, and it remains unknown if predictions about one feature affect predictions about another feature, and whether this plausible interaction affects model precision and the actual learning. A neurophysiological marker of model precision is the mismatch negativity (MMN), which is an electrical brain response to a deviant auditory stimulus among standards [8]. Here, we used the MMN to investigate how predictions based on statistical learning are affected when predictability of stimulus-onset is manipulated by using random stimulus-onset asynchronies

**Competing interests:** The authors have declared that no competing interests exist.

(SOAs). In a traditional auditory oddball paradigm with isochronous stimulation, the repeated presentation of standards dispels uncertainty about the stimuli (i.e. an increase of the precision of predictions) in terms of their onset and content. This results in a reduction in event-related potentials (ERPs) to standards and to the elicitation of an error-signal (the MMN) in response to deviants [9, 10]. On the other hand, when stimuli are presented non-isochronously (e.g., random SOAs), the establishment of first-order predictions about stimulus-onset becomes less precise because the suppression of prediction errors is less efficient [9]. Thus, the cost of reduced precision is reflected in smaller error-signal responses to deviants.

Does the same rationale hold for prediction error signals in statistical learning paradigms and in artificial grammar learning paradigms? Statistical learning is a key mechanism for the detection of regularities in our environment and has become a major field in cognitive psychology [11–13]. Statistical learning studies typically use isochronous presentation of stimuli, i.e. stimuli with predictable stimulus-onset (and, to our knowledge, no previous statistical learning study has used unpredictable stimulus-onsets). Similarly, in the field of artificial grammar learning, stimulus-onsets are typically predictable (usually isochronous; for implicit learning of rhythm or temporal patterns see [14, 15]). Indeed, from the perspective of formal language theory [16, 17] it should not make a difference whether stimuli of an artificial grammar learning study are presented in an isochronous fashion or not, because the rules of the grammar are the same (given that stimulus duration is not part of the grammar). Likewise, in a statistical learning study, it should not make a difference whether stimuli are presented isochronously, because the transitional probabilities of the events are identical (given that stimulus duration is independent of transitional probabilities). Therefore, in computational modelling of implicit learning (e.g., [18]), SOAs can theoretically be disregarded given that they are not part of the regularities underlying the simulated learning process. In that respect, the present study may shed light on plausible interaction effects of learning and temporal uncertainty of stimuli [4].

Predictability in terms of stimulus-onset has also been manipulated in oddball studies using the classical MMN, but the results are not consistent. A number of studies suggest that the MMN amplitude is decreased when random SOAs are used [19–21]. Tavano et al. [22] reported that random SOAs decreased the MMN amplitude only when a deviant was repeated in a unpredictable fashion, suggesting that temporal regularity is crucial only for predictions regarding deviant repetition probability. However, other studies found no effect on the MMN amplitude when stimuli were presented in random temporal structure compared to isochronous structure [23], or when the SOA was manipulated [24]. Therefore, more evidence is needed to clarify whether auditory sensory memory processes, underlying the generation of the MMN, are affected by manipulation of predictability when using randomly varied SOAs.

To investigate how statistical learning and auditory deviance detection are affected by temporal predictability we employed a variation of an experimental paradigm used in a previous study [25]. We constructed a continuous auditory stream of sound triplets with deviants that were either (a) statistical, in terms of transitional probability, (b) physical, due to location change ("standards" were presented from one direction, whilst "deviants" were presented from the other direction) or (c) double deviants, i.e. a combination of the two (see Fig 1). Statistical and physical deviants tapped different stimulus dimensions. Specifically, statistical deviants regarded the stimuli content, or the "what", whilst the physical deviants regarded the stimuli location, or the "where". Contrary to our previous study [25], where a constant SOA was used, in the current study we used random SOAs as a means to manipulate the predictability of stimulus-onset. In that way, we manipulated a third stimulus dimension which was the time, or the "when".

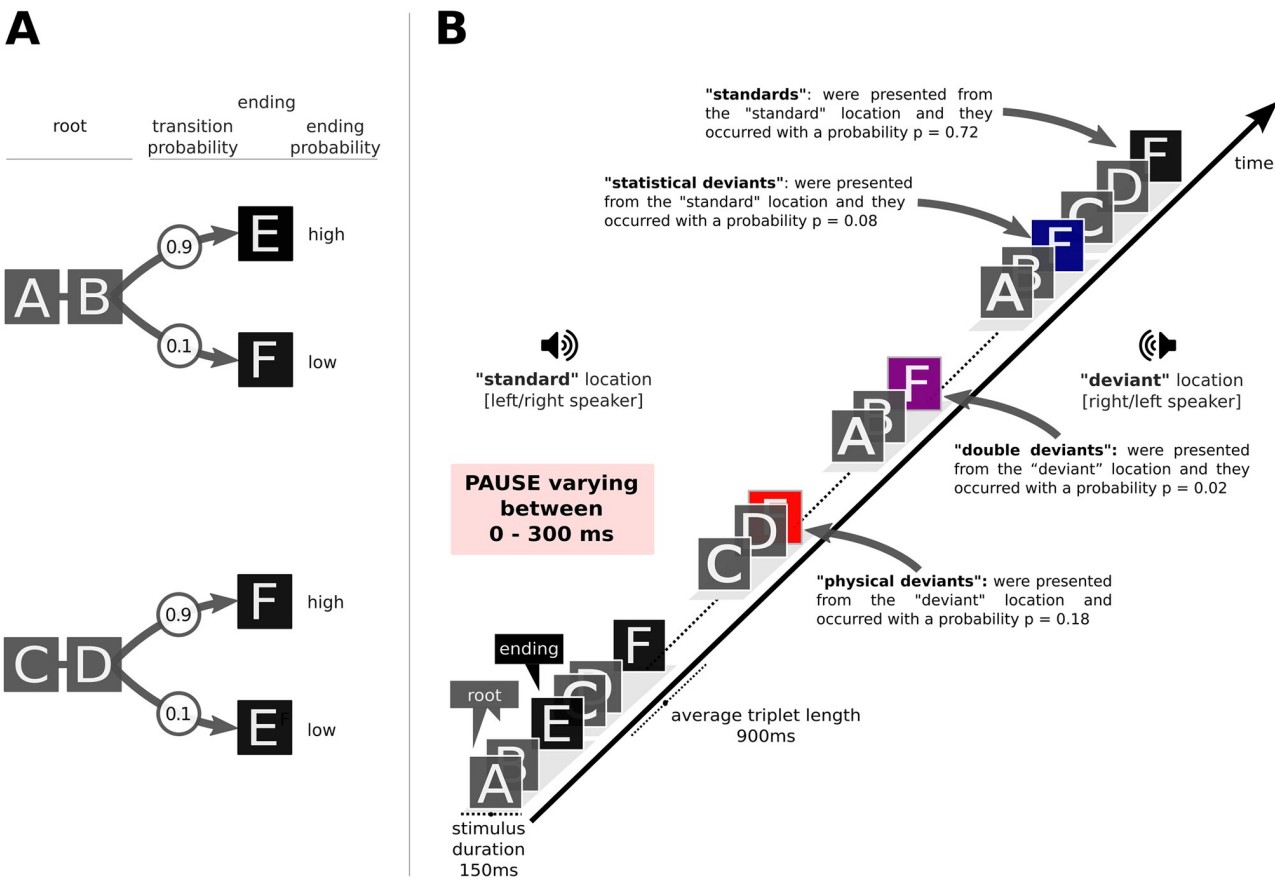

**Fig 1. The experimental paradigm.** (A) The four triplets generated from the 6 sounds. The letters A to E are used to refer to the sounds. The first two items of the triplet are referred to as "root" and the last item as "ending". Statistical deviants were created by varying the transition probability from root to ending within two levels, high (p = 0.9) and low (p = 0.1). Triplet roots (AB or CD) are occurring with a constant transitional probability (p = 0.5) after any of the triplet endings (E or F). **(B)** The auditory stream of pseudorandomly concatenated triplets with standard ending triplets (letter in blackbox), statistical deviant ending triplets (letter in blue box), physical deviants (letter in red box) and double deviants (letters in purple box). Physical deviants were generated by switching speaker, leading to a spatial location change of 60˚ in the azimuth plane.

Our aim was to examine how effects of predictability are reflected in two ERP components, the statistical MMN (sMMN) [25, 26] and the location MMN. The sMMN was used as a neurophysiological marker of statistical learning and the location MMN as a marker of auditory deviance detection. We tested whether statistical or physical deviants would elicit an sMMN or a location MMN respectively, even when SOAs are random. A secondary scope of our study was to compare the results from the current study (using random SOAs) and our previous experiment using isochronous SOAs [25]. We expected that both sMMN and location MMN would be affected by the predictability of the SOA, and therefore we tested for an interaction between the ERP-effects elicited by deviant events (deviants compared to standards) and timing (isochronous vs. non-isochronous). Within the PC framework, unpredictability (computed as Shannon entropy) expresses the level of uncertainty about upcoming events [27]. For instance, low entropy is indicative of highly upcoming events [6]. Compared to our previous study, in the current paradigm the uncertainty or the entropy of the "when" changed from minimal (isochronous events with no uncertainty) to maximal (random SOAs with high uncertainty [28, 29]), whereas the entropy in terms of the "what" (what tones are presented) and "where" (sound direction) remained the same. Therefore, we expected that

manipulating the entropy of the "when" (by switching from isochrony with zero entropy in the temporal structure, to randomly varied SOAs with maximal entropy) would impact the predictive processes of "what" and "where". In PC terms, this manipulation (switching to random SOAs) reduced precision of the predictive model, which we expected to be reflected in a reduction of the amplitudes of error signals (the sMMN and the location MMN). By doing so, our study also aimed at adding evidence to the existing theoretical frameworks describing the cognitive processes during sequence learning. For instance, computational models of implicit learning neglect stimulus timing because sequential models (e.g., n-grams or Markov models) focus on the ordering of the events [16, 17]. On the other hand, PC suggests that prediction-driven learning relies on the estimation of probabilities [30], but it remains unclear whether modelling of one stimulus dimension (e.g., pitch) interacts with another stimulus dimension (e.g., timing). Deheane et al. [31] have proposed a theoretical framework for sequence learning with increasing levels of abstraction. Interestingly, transitions and timing knowledge are placed at the first level, suggesting the tight interconnection between the learning of the "what" and the "when". This supports our hypothesis that unpredictable (non-isochronous) stimulus onsets would impede accurate predictions, and thus impair learning as reflected in the amplitudes of the statistical and the location MMN.

## Materials and methods

### Participants

Datasets from 21 adults (11 women; mean age = 24.10 years, $SD$ = 6.21) were included in the analysis. Exclusion criteria were hearing impairments, history of neurological disease, and musical training for more than 2 years beyond regular school lessons (according to self-report). All participants were compensated financially. Participants provided written informed consent before the experiment. The study was carried out in accordance with the guidelines of the Declaration of Helsinki, and approved by the Regional Committee for Medical and Health Research Ethics for Western Norway (Reference Number: 2018/2409). The 21 datasets from the current experiment were compared with 21 datasets (12 women; mean age = 22.43 years, $SD$ = 2.39) from a previous experiment [25]. See Table 1 for a comparative view of demographic information between the two experiments.

### Stimuli

**Sound triplets.**  The experimental paradigm is a variation of a previous study [25]. Six sounds were created and each sound was a combination of a Shepard tone and a percussion sound. Shepard tones [32] were employed in order to reduce any percept of pitch along with any auditory grouping based on pitch. The use of pure tones was rejected during pilot testing because emergent Gestalt formations (i.e., ascending or descending triads) were confounding learning of the triplet structure [33, 34]. The six Shepard tones were computer-generated complex tones over the following frequencies (F3: 174.61 Hz, G3: 196.00 Hz, A3: 220.00 Hz, B3: 246.94 Hz, C#4: 277.18 Hz and D#4: 311.13 Hz). The particular feature of Shepard tones is that although they differ in frequency, they are ambiguous when it comes to judging the relative pitch. Therefore, the six Shepard tones would sound differently, but the perceived direction of

**Table 1. Comparative view of demographic information between the two experiments.**

|  | Non-isochronous stimulation (current study) | Isochronous stimulation (Tsogli et al. 2019) |
| --- | --- | --- |
| Sample size | 21 (11 women) | 21 (12 women) |
| Mean age | 24.10 (SD = 6.21) | 22.43 (SD = 2.39) |

the tones would be ambiguous for the participants (i.e., whether the tones were going "up" or "down"). Each sound had a duration of 150 ms (compared to 220 ms in the previous experiment; [25]), including a fade in of 10 ms and a fade out of 20 ms. To manipulate stimulus-onset predictability we delivered the stimuli in a non-isochronous fashion; a randomly varying pause between 0 and 300 ms followed each sound. Therefore, the stimulus-onset asynchrony (SOA) varied randomly between 150 and 450 ms, with an average SOA of 300 ms (identical to the SOA in [25]). See Table 2 for a comparative view for the details of the experiment design for the two studies.

The letters A to F will be used to refer to the six sounds which were arranged into four different triplets (ABE, ABF, CDF and CDE; see Fig 1A). Sounds A to D were used for the first two sounds of the triplets or the "triplet roots" (AB and CD). The remaining two sounds (E and F) were used as "triplet endings". The stimuli arrangement of the current paradigm represents a 1st-order Markov model or bigram model with strictly 2-local distribution [16] and can be placed in the "subregular" hierarchy within the extended Chomsky hierarchy [35] or in the "regular" class within the classical Chomsky hierarchy. Chomsky [36] proposed a nested hierarchy of classes with increasing complexity as a way to formalise the increasing complexity in the structure of natural languages. Subregular structures are less complex compared to regular ones and the extended Chomsky hierarchy accounts for this difference by placing the two types of structures in different classes.

An additional set of six sounds was created for the practice trials before the experiment. These sounds were created similar to the sounds of the experiment, but differed in the frequency of the Shepard tones (E3: 164.81 Hz, F#3: 184.99 Hz, G#3: 207.65 Hz, A#3: 233.08 Hz, C4: 261.62 Hz and D4: 293.66 Hz) and in the percussive sounds that were used (woodblock, tambourine, agogo bells, castanet, hi-hat and bass drum). Finally, an additional higher-pitched sound was created (C#5: 554.37 Hz, not combined with a percussive sound) to serve as target sound for the cover task that participants had during practice trials and the experiment.

**Table 2. Comparative view of experiment design details for the two experiments.** Interstimulus interval denotes the silent interval between the offset of one tone and the onset of the next one. SOAs denote the interval between the onset of one tone and the onset of the next one.

| | Non-isochronous (current study) | Isochronous stimulation (Tsogli et al. 2019) |
|---|---|---|
| Sounds (Shepard tone & percussive sound) | F3: 174.61 Hz & surdo | |
| | G3: 196.00 Hz & tambourine | |
| | A3: 220.00 Hz & agogo bells | |
| | B3: 246.94 Hz & hi-hat | |
| | C#4: 277.18 Hz & castanet | |
| | D#4: 311.13 Hz & woodblock | |
| Stimuli duration | 150 ms | 220 ms |
| Target sound | C#5: 554.37 Hz | |
| Interstimulus interval | randomly varying between 0 and 300 ms | 80 ms |
| SOAs | randomly varying between 150 and 450 ms | 300 ms |
| Probabilities of triplet endings | Standards: p = 0.72 | |
| | Statistical deviants: p = 0.08 | |
| | Physical deviants: p = 0.18 | |
| | Double deviants: p = 0.02 | |
| Triplet stream | 6 blocks of 7 min | |

Importantly, the arrangement of sounds (A to F) was permuted across participants as a way to guarantee that possible acoustical differences between sounds would not bias the brain responses of interest.

**Stimulus location.** In order to generate physical deviants, the location of the sound stimuli was manipulated, featuring a spatial location change of 60˚ angle in the azimuthal plane: If the standard stimuli were presented from the direction of one speaker, the physical deviants were presented from the other one. The stimuli location was arranged as follows: Each of the sounds of the triplet root (A to D) was presented with a probability p = 0.95 from the "standard" and with a probability p = 0.05 from the "deviant" location. The triplet ending was presented with a probability p = 0.80 from the "standard" side, and with a probability p = 0.20 from the "deviant" side. Only triplet endings were evaluated when assessing physical deviance. "Standard" and "deviant" location was balanced across blocks and counterbalanced between participants whether they would have left or right as preferential (standard) direction for the first block.

**Triplet endings.** Four categories of triplet endings were formed by manipulating the transition probability of the triplet ending from high probability (p = 0.90) to low probability (p = 0.10) and the sound location from "standard" (p = 0.80) to "deviant" location (p = 0.20). The four categories were as follows:

1. Standards: featured a high transition probability (p = 0.90) and were presented from the "standard" location (p = 0.80). Thus, they occurred with a probability of p = 0.72.

2. Statistical deviants: featured a low transition probability (p = 0.10) and were presented from the "standard" location (p = 0.80). Thus, they occurred with a probability of p = 0.08.

3. Physical deviants: featured a high transition probability (p = 0.90) and were presented from the "deviant" location (p = 0.20). Thus, they occurred with a probability of p = 0.18.

4. Double deviants: featured a low transition probability (p = 0.10) and were presented from the "deviant" location (p = 0.20). Thus, they occurred with a probability of p = 0.02.

**Triplet streams.** Four hundred triplets were pseudorandomly concatenated into six pause-free streams (referred as "blocks") of about 7 min duration each. Triplets were presented in a pseudorandom order so that triplets from the low probability set were separated by at least three triplets from another set. Triplet roots (AB or CD) followed any of the two triplet endings (E or F) with a constant transitional probability (TP = 0.5). So, for example ABE could be followed by either ABE, CDF, ABF or CDE.

**Information content and conditional entropy values.** In the current paradigm the values of information content and conditional entropy are indicative of the level of predictability or uncertainty (see Table 3). Specifically, the information content of triplet endings decreases in analogous manner to the probability of occurrence. For instance, standard endings are characterised by the lowest information content or surprisal (IC = 0.15) whereas the information content of statistical deviants is higher (IC = 3.32) because they are less frequent and therefore are expected to induce greater surprise. Within the larger setting of information theory and PC, "surprisal" is suggested to express the surprise due to a mismatch between the sensory signals and those predicted in a formal manner [7]. Information content and conditional entropy were calculated based on the Eqs 1 and 2 [37]. In the current paradigm the conditional entropy at the triplet ending is relatively low (H = 0.46). Thus, an interesting aspect of this paradigm is the presentation of an unpredicted event in a position with low expected surprise. While the predictability in regard to the "what" and "where" is identical to the previous experiment [25]

**Table 3. Predictability values for triplet endings.**

| Predicting the "what" | |
|---|---|
| **Information Content** | |
| Standard ending (p = 0.9) | Statistical deviant (p = 0.1) |
| 0.15 | 3.32 |
| **Conditional Entropy of the ending** | |
| 0.46 | |
| **Predicting the "where"** | |
| **Information Content** | |
| Standard location (p = 0.8) | Deviant location (p = 0.2) |
| 0.32 | 2.32 |
| **Predicting the "when"** | |
| **Entropy in isochronous mode** | **Entropy in non-isochronous mode** |
| 0 | 8.23 |

the predictability in regard to the "when" has changed. An isochronous temporal structure, from an information theory perspective, is characterised by zero entropy and thus low uncertainty [29], contrary to a non-isochronous structure which features increased entropy and uncertainty. In the current paradigm the entropy of the irregular temporal structure was calculated on the basis that the pause between the stimuli varied randomly between 0 and 300 ms, therefore, the set of pauses comprised of 301 equiprobable elements with p = 1/301, which results in a entropy of 8.23 bits during non-isochronous stimulation, according Eq (3).

$$h(ending) = log\left(\frac{1}{P(ending)}\right) \tag{1}$$

The conditional entropy at the triplet ending, for all possible triplet endings, was calculated based on Eq (2):

$$H(ending|root) \equiv \sum P(ending|root)log\left(\frac{1}{P(ending|root)}\right) \tag{2}$$

The entropy of the irregular temporal structure was calculated based on Eq (3):

$$H(ISIs) \equiv \sum P(ISI)log\left(\frac{1}{P(ISI)}\right) \tag{3}$$

## Procedure

The experiment took place inside an electro-magnetically shielded chamber. Participants sat in a chair in front of a desk with a monitor. Their seating position was chosen so that it formed an equal side triangle with the speakers and that their eyes were at the level of the centre of the screen. Prior to the main experiment, participants were asked to complete a discrimination test which will be described later.

After the completion of the discrimination test, the main experiment started with a set of instructions displayed on the monitor. When necessary were additional instructions given orally by the experimenter. The experiment consisted of 6 blocks, each one comprising an exposition phase of about 7 min and each followed by a behavioural task of about 2 min resulting in a total duration of about 1 hour (including pauses). During the exposition phase, the auditory stimuli were presented via the speakers while participants were asked to watch a silent

movie on the monitor in front of them. The silent movie was a documentary about birds which was selected because it was expected to be neither too arousing nor cognitively too demanding and there was no presentation of letters or people speaking. Participants were not informed about the regularities in the arrangement of the stimuli, to ensure that any kind of learning throughout the experiment was implicit. To ensure that participants were attentive to the stimuli, a cover task was used: The participants were asked to press the space bar every time they heard the (higher-pitched) target sound. There were examples of the target sound in the instructions, followed by practice trials (lasting about 1 min) containing a relatively high number of target sounds. The practice trials were repeated if participants did not detect at least 80% target sounds (or had a too large number of false alarms). At the end of each of the 6 blocks, there was a behavioural test which will be described later.

The electroencephalogram (EEG) was recorded during the whole experiment. Participants were asked to avoid movement, especially of jaws and eyes, in order to minimize artifacts in the EEG recording. The experimenter was present in an adjacent room throughout the experiment. He could monitor the participant's state at all times with a camera directed at the participant's head. Participants were instructed to give a sign to the camera should they need to interrupt the experiment. At the end of the experiment, participants were inquired about their awareness in regard to the stimuli structure. They were asked to explicitly state any rule or pattern they were able to detect in the stimuli they had heard.

## Discrimination test for acoustical perception of sounds prior to the main experiment

Participants responded to a discrimination test prior to the main experiment. The purpose of the test was to ensure that the stimuli that would be presented during the main experiment were acoustically distinguishable in terms of pitch despite the shorter duration of the sounds compared to the one used in the previous experiment [25].

The test consisted of twelve trials. In each trial, participants heard two sound sequences with 1 sec silence gap between the sequences. Each sequence comprised of three sounds. The sequences were either the same or different in terms of the tones that were presented. Participants were asked to press "1" if they felt that the sequences sounded the same or "2" if they were different. The discrimination test started with practice trials. The participant could proceed to the main test only if he had answered correctly three trials. The order of the trials was randomized between subjects.

For the discrimination test the same sounds as for the main experiment were used. The sounds were combined into triplets (such as AEF, BAF, FEB, EDC etc.) which were different from those presented in the main experiment (see Fig 1A). The interstimulus interval was 50 ms. The arrangement of the sounds (A to F) was permuted across participants.

**Familiarity test and confidence rating during the main experiment.** At the end of each block of the main experiment, a behavioural test (two-alternative, forced-choice goodness-of-fit rating) assessed whether participants became familiarised with the statistical regularities in the arrangement of the stimuli. The test consisted of 12 trials. In each trial participants heard two triplets with either high or low probability ending and were asked to choose which sequence sounded more familiar to them (pressing either "1" for the first or "2" for the second sequence). The sequences ending with high probability sounds occurred more often during the exposition phase and were regarded as the correct ones in terms of familiarity. Afterwards, the participants rated their level of confidence about their choice (pressing from "1" for absolutely unsure to "5" for absolutely certain).

During the familiarity test all four possible triplet combinations (ABE vs. ABF, ABF vs. ABE, CDF vs. CDE, or CDE vs. CDF, see Fig 1) were repeated three times in random order. All sequences were presented binaurally, none contained a location change, and there was a pause of 335 ms between the triplets. Consecutive trials did not use the same triplet root and the order of presentation of the endings was counterbalanced.

## Data recording and analysis

**EEG recording.** The EEG signal was recorded from 59 passive electrodes mounted in an EEG cap, according to the international 10–10 system and at a sampling rate of 500 Hz using BrainAmps DC (Brain Products GmbH, Munich, Germany). Additional electrodes were placed: (a) on each mastoid, while the left served as reference during recording, (b) one on the back of the neck as ground, (c) two at the outer part of the canthi of both eyes to record the horizontal electrooculogram (EOG), and (d) two below and above the right eye to record the vertical EOG. All electrode impedances were kept below 5 kΩ.

**Processing of EEG data.** EEG data were analysed using EEGLAB 13 [38] within MATLAB® R2016b (The MathWorks Inc., Natick, MA). EEG data recorded during the behavioural part of the experiment (at the end of every block) were not evaluated. Visual inspection of the EEG data was performed to reject any periods with excessive artifacts or any faulty channels. An Independent Components Analysis was conducted to remove eye and muscular artifacts. Subsequently, EEG data were re-referenced to the algebraic mean of the left and right mastoid electrodes and filtered using a 30 Hz low-pass filter (2750 points, finite impulse response, Blackman).

Samples were rejected whenever the standard deviation within a 200 or 800 ms gliding window exceeded 25 $\mu$V at any electrode channel (including the EOG channels). Afterwards, data were epoched, excluding epochs following acoustical deviants or button presses (within 3 secs; i.e., rejecting activity related to the cover task of the participant). Additionally, to ensure cleaner EEG data over the MMN window, only epochs with at least 200 ms from the adjacent trigger were included in the analysis. In that way we ensured that only tones with at least 50 ms apart from the adjacent one (either before or after) would be included in the analysis and thus reduce any possible contamination of ERP responses from adjacent sounds occurring too early. ERPs were calculated for low and high probability triplet endings with or without location change of the sound, from -100 to 400 ms relative to stimulus-onset and using a 100 ms pre-stimulus baseline correction.

For statistical evaluation, electrodes were clustered into nine regions of interest (ROIs), namely frontal left (F7, F5, F3, FT7, FC5, FC3), frontal middle (F1, FZ, F2, FC1, FCZ, FC2), frontal right (F8, F6, F4, FT8, FC6, FC4), central left (T7, C5, C3, TP7, CP5, CP3), central middle (C1, CZ, C2, CPZ), central right (T8, C6, C4, TP8, CP6, CP4), parietal left (P7, P5, P3, PO7, PO3, O1), parietal middle (P1, PZ, P2, POZ, OZ) and parietal right (P8, P6, P4, PO8, PO4, O2). The time windows for statistical analysis were selected in accordance with previous studies (see Introduction) and based upon visual inspection.

**Statistical analyses.** Statistical analyses were conducted using JASP (JASP Team, 2019, version 0.11). For the behavioural data recorded during the discrimination test, the cover task and the familiarity test, the mean percentage of correct responses was calculated. Only for the familiarity test, the mean score was subsequently compared against chance level (0.5; one sample t-test, $\alpha = 0.05$). The neurophysiological data were analysed using both frequentist and Bayesian statistics within JASP [39]. Repeated measures analyses of variance (ANOVAs) were carried out to assess the ERPs of the current experiment with random SOAs and then compare the results between the current experiment and the previous one where a constant SOA was

**Table 4. Comparative view of methodological details for the two experiments.**

|  | Non-isochronous (current study) | Isochronous stimulation (Tsogli et al. 2019) |
|---|---|---|
| Processing of EEG data | EEGLAB 13, MATLAB R2016b | |
| Statistical analyses | JASP 0.11 | SPSS 25 |
| Procedure | Discrimination test | |
|  | Practice | Practice |
|  | Main experiment | Main experiment |
| Behavioural data | Discrimination test | |
|  | Cover task during exp. phase | Cover task during exp. phase |
|  | Familiarity test | Familiarity test |
| Statistical analyses | Bayesian ANOVA | |
|  | Frequentist ANOVA | Frequentist ANOVA |
| Statistical MMN | | |
| Time window | 180 to 260 ms | 180 to 260 ms |
|  | 150 to 200 ms | |
| Within subject factors | transition probability | |
|  | scalp area | |
|  | lateralisation | |
|  | block | |
| Location MMN | | |
| Time window | 150 to 220 ms | 150 to 220 ms |
| Within subject factors | physical deviance | |
|  | scalp area | |
|  | lateralisation | |
|  | block | |

used. Specifically we examined: (1) the ERPs of statistical deviants (sMMN) under non-isochronous stimulation (2) the ERPs to physical deviants (location MMN) under non-isochronous stimulation, (3) the interaction of the isochronicity with the sMMN which would entail a comparison between the results of the current (random SOAs) and the previous experiment (constant SOA, [25]) and (4) the interaction of the isochronicity with the location MMN which would entail a comparison between the results of the current (random SOAs) and the previous experiment (constant SOA, [25]). ANOVAs were conducted by including the following factors: (a) scalp area (frontal, central, posterior), (b) lateralisation (left, midline, right) and (c) experiment block (first, second, third; the first, middle and last two blocks of the 6 blocks of the experiment were grouped in order to obtain a better signal-to-noise-ratio). ERPs for the sMMN were assessed over two time windows, the established sMMN time window 180—260 ms [25] and an earlier time window 150—200 ms. ERPs for the location MMN were assessed over the time window 150—220 ms. See Table 4 for a comparative view of the methodological details for both experiments.

## Results

### Behavioural data

Prior to the main experiment a discrimination test was conducted to guarantee that the stimuli were acoustically distinguishable in terms of pitch (see Methods). Participants correctly differentiated sound sequences that were of the same or different pitches with an average score of 95.6% ($SEM$ = 1.24%). Thus, all participants could discriminate the stimuli well.

**Cover task during the exposition phase.**    As mentioned earlier, participants were not informed about the statistical regularities underlying the stimuli presented during the exposition phase, but instead were provided with a cover task, namely to detect a (higher-pitched) target sound. Participants detected on average 97.9% of these target sounds ($SEM = 0.85\%$), indicating that participants attended the sounds although they were watching a silent movie.

**Familiarity test.**    At the end of each exposition block, a familiarity test was presented to test whether participants had learned the underlying regularities of the stimuli. It was expected that participants would classify the sequences ending with high probability sounds as more familiar compared with those ending with low probability sounds (see Methods). Participants achieved an overall score of 50.6% ($SEM = 2.00\%$) in classifying the sequences ending with high probability sounds correctly as more familiar. The performance was not different from chance level ($p = 0.77$, Cohen's $d = 0.065$) indicating that participants remained unaware of the regularities governing the arrangement of the stimuli. During the debriefing session no participant showed awareness of the underlying stimuli structure.

## ERPs of triplet endings

First, we will present the ERPs of statistical and physical deviants under the non-isochronous stimulation of the current experiment. Subsequently, we will present the interaction of the isochronicity with the statistical and location MMN which entails a comparison between the results of the current experiment (i.e., where random SOAs were used) and our previous study [25] where stimuli were presented isochronously (i.e., where a constant SOA was used).

**ERPs of statistical deviants (sMMN) under non-isochronous stimulation.**    Fig 2A shows ERPs elicited by standards and statistical deviants (high and low probability triplet endings) under non-isochronous stimulation with low temporal predictability. Although it seems that statistical deviants, compared to standards, elicited a tiny sMMN at around 190 ms, this difference was statistically not significant. A GLM ANOVA for repeated measurements with the within-subject factors transition probability (high vs. low), scalp area (anterior, central and posterior), lateralisation (left, midline and right) and block (1 to 3) for the time window from 180 to 260 ms after the onset of the triplet ending showed no significant effect of transition probability ($p = 0.80$, $\eta^2 < 0.001$), no significant effect of block ($p = 0.61$, $\eta^2 = 0.004$) and no interaction between transition probability and block ($p = 0.63$, $\eta^2 = 0.004$).

To assess whether or not the current data support the hypothesis that low probability events elicit an sMMN, a Bayesian ANOVA for repeated measurements was conducted with identical factors as for the GLM ANOVA and an identical time window. The normality of the residuals was assessed. For transition probability, a Bayes factor of $BF_{01} = 1.254e^{10}$ (error = 3.81%) provided substantially stronger evidence in favour of the null hypothesis, i.e., the assumption that standards and statistical deviants did not elicit different brain responses (i.e., no amplitude difference in the assessed time window from 180 to 260 ms).

Therefore, the current data are more likely not to represent an sMMN elicitation. To ensure that the present finding of no effect of triplet ending is not simply due to the choice of the time window from 180 to 260 ms (which was based on our previous study; [25]), the analysis was repeated over a tighter time window from 150 to 200 ms, during which the difference between the waveforms appeared to be largest. The normality of the residuals was assessed prior to the analysis. Again, the Bayesian ANOVA showed no evidence for any effect of transition probability, $BF_{01} = 1.254e^8$ indicating that the current data by a factor of $1.254e^8$ are more likely to be consistent with the null hypothesis, indicating no elicitation of an sMMN.

Triplet endings also elicited an early negativity with a latency of 50 ms which was followed by a positivity with a latency of 130 ms. The early negativity was assessed with a GLM ANOVA

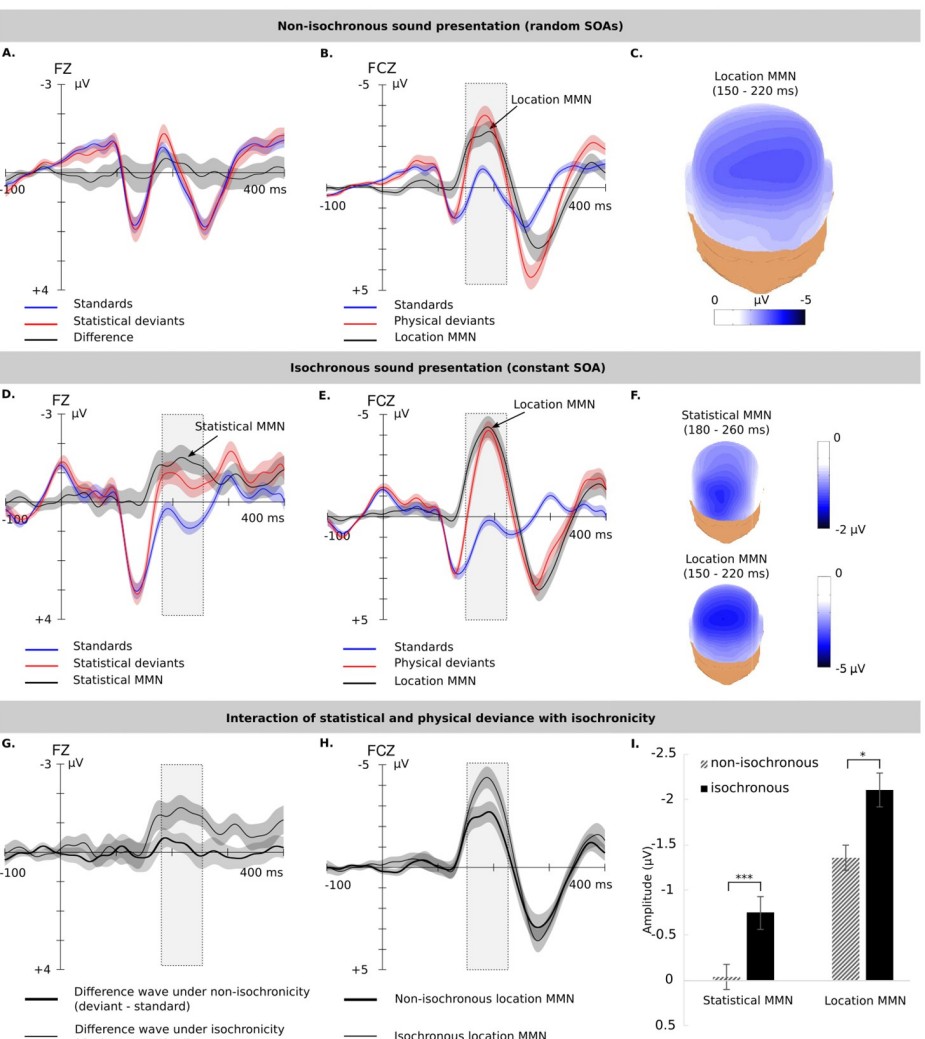

**Fig 2. ERPs of statistical and physical deviants as recorded in the current study with non-isochronous sound presentation, and in the previous study [25] with isochronous sound presentation.** (A) Mean ERP waves for standards and statistical deviants and the difference (no sMMN) under non-isochronous as recorded at electrode Fz. The shaded area on the ERPs represents the SEM. **(B)** Mean ERP waves for standards and physical deviants, as recorded from the electrode FCz. The dotted rectangle indicates the time-window for statistical analysis (150–220 ms). **(C)** Isopotential map showing the scalp distribution of the location MMN over the time-window (150–220 ms). **(D)** The sMMN under isochronous (high temporal predictability) stimulation as captured at electrode Fz. The dotted rectangle indicates the time-window for statistical analysis (180–260 ms). **(E)** The location MMN under isochronous stimulation as captured at electrode FCz. **(F)** Isopotential maps showing the scalp distribution of the sMMN over the window (180–260 ms) and the location MMN over the window (150–220 ms). **(G)** Separate traces for the differences to statistical deviants under isochronous and non-isochronous stimulation as captured at electrode Fz. **(H)** Separate traces for the location MMN under isochronous and non-isochronous stimulation as captured at electrode FCz. **(I)** Both sMMN and location MMN amplitudes decline significantly under non-isochronous stimulation.

for repeated measurements with the within-subject factors transition probability (high vs. low), scalp area (anterior, central and posterior) and lateralisation (left, midline and right) for the time window from 0 to 100 ms after the onset of the triplet ending which showed no effect of probability ($p = 0.43$, $\eta^2 = 0.003$). The lack of difference was in agreement with the Bayesian ANOVA with identical factors as for the GLM ANOVA, indicating that the current data by a factor of 9.225e$^{24}$ are more supportive of the null hypothesis that there most likely was no effect

of probability. Likewise, the analysis for the positivity with identical factors for the time window from 100 to 200 ms showed not effect of probability ($p = 0.71$, $\eta^2 = 0.001$) and in a similar vein the Bayesian ANOVA indicated that the current data by a factor of $1.965e^4$ more likely do not show an effect of probability. The normality of the residuals was assessed prior to the analysis.

**ERPs of physical deviance (location MMN) under non-isochronous stimulation.**
Fig 2B shows the location MMN elicited under non-isochronous stimulation (low temporal predictability) by high probability endings with and without physical deviance and was maximal at around 170 ms. A GLM ANOVA for repeated measurements with the within-subject factors physical deviance (with vs without location change), scalp area (anterior, central and posterior), lateralisation (left, midline and right) and block (1 to 3) for the time window from 150 to 220 ms after the onset of the triplet ending showed a significant effect of physical deviance ($F(1, 20) = 69.30$, $p < 0.001$, $\eta^2 = 0.265$) and block ($F(2, 40) = 3.67$, $p < 0.05$, $\eta^2 = 0.012$). Significant effects were also found for scalp area ($F(2, 40) = 18.97$, $p < 0.001$, $\eta^2 = 0.043$), lateralisation ($F(2, 40) = 8.13$, $p = 0.001$, $\eta^2 = 0.009$) and significant interactions for: (1) physical deviance and scalp area (reflecting that the effect was largest over the frontal scalp areas; $F(2, 40) = 38.33$, $p < 0.001$, $\eta^2 = 0.042$), (2) physical deviance and lateralisation (reflecting that the effect was greater over the midline with slight left lateralisation, see Fig 2C; $F(2, 40) = 11.93$, $p < 0.001$, $\eta^2 = 0.005$) and (3) physical deviance, scalp area and lateralisation (showing the effect was largest over the middle frontal scalp sites and slightly left-lateralised ($F(4, 80) = 3.07$, $p < 0.05$, $\eta^2 = 0.001$).

Triplet endings also elicited an early negativity with a latency of 50 ms (N50). The early negativity was assessed with a GLM ANOVA for repeated measurements with the within-subject factors physical deviance (with vs. without location change), scalp area (anterior, central and posterior) and lateralisation (left, midline and right) over the time window from 0 to 100 ms which showed no effect of probability ($p = 0.888$, $\eta^2 < 0.001$). Likewise, the Bayesian ANOVA with identical factors as for the GLM ANOVA, showed that by a factor of $1.295e^{44}$ the data did not represent an effect of probability. The normality of the residuals was assessed prior to the analysis.

**Interaction of statistical deviance and isochronicity.** To investigate whether the mode of stimulation (i.e., isochronous or non-isochronous) interacted with the elicitation of the sMMN, a comparison between the results of the previous study [25] with a constant SOA at 300 ms (see Fig 2D) and the results of the current experiment with random SOAs was carried out. Specifically a repeated measures GLM ANOVA was conducted with within-subjects factors: (1) transition probability (high vs low probability triplet ending), (2) scalp area (anterior, central and posterior), (3) lateralisation (left, midline and right) and (4) block (1 to 3) and with one between-subjects factor the isochronicity (isochronous vs non-isochronous), for the time window from 180 to 260 ms after the onset of the triplet ending. The analysis showed a significant effect of probability ($F(1, 40) = 15.32$, $p < 0.001$, $\eta^2 = 0.032$) and a significant interaction of transition probability and isochronicity ($F(1, 40) = 12.41$, $p = 0.001$, $\eta^2 = 0.026$) reflecting that the effect of statistical deviance was significant under isochronous, but not under non-isochronous stimulation (see Fig 2A, 2D and 2G). The observed interaction between transition probability and isochronicity, further demonstrates the effect of the uncertainty (entropy) in regard to the timing of the events; switching from an auditory input with low uncertainty (Entropy = 0) to an input with high uncertainty (Entropy = 8.23; see Table 3) diminished significantly the effect transition probability (see Fig 2I).

To ensure that the finding of an interaction between the transition probability and the isochronicity is not due to the choice of the time window from 180 to 260 ms (which was based on our previous study; [25]), the same analysis was repeated over the time window from 150 to

200 ms. Again, the analysis showed significant effects of probability ($F(1, 40) = 8.40$, $p < 0.01$, $\eta^2 = 0.017$), of block ($F(2, 80) = 4.89$, $p < 0.05$, $\eta^2 = 0.013$) and a significant interaction of probability and isochronicity ($F(1, 40) = 8.22$, $p < 0.05$, $\eta^2 = 0.016$).

**Interaction of physical deviance and isochronicity.**   In the same way the effect of isochronicity to the physical deviance was examined by comparing the results of the current experiment with random SOAs and the previous one with a constant SOA of 300 ms [25] (see Fig 2E and 2F). A GLM ANOVA for repeated measurements was conducted with the within-subjects factors: (1) physical deviance (high-probability endings with and without location change), (2) scalp area (anterior, central and posterior), (3) lateralisation (left, midline and right) and (4) block (1 to 3) and the between-subjects factor mode (isochronous vs non-isochronous), for the time window from 150 to 220 ms after the onset of the triplet ending. The analysis revealed a significant effect of physical deviance ($F(1, 40) = 150.11$, $p < 0.001$, $\eta^2 = 0.338$), a significant effect of block ($F(2, 80) = 10.25$, $p < 0.001$, $\eta^2 = 0.013$) and a significant interaction between physical deviance and isochronicity ($F(1, 40) = 6.96$, $p < 0.05$, $\eta^2 = 0.016$) reflecting that the effect of physical deviance diminishes significantly under non-isochronous stimulation (see Fig 2B, 2E and 2H). Similarly to the statistical deviance, switching from an auditory input with low uncertainty (Entropy = 0) to an input with high uncertainty (Entropy = 8.23; see Table 3) diminished significantly the effect physical deviance (see Fig 2I).

## Discussion

The aim of the current study was to examine whether the reduced predictability due to random SOAs affects the sMMN and the location MMN within a statistical learning paradigm. Neurophysiological and behavioural responses were assessed for evidence indicating that participants would track the statistical regularities and thus optimize the predictive model. It was expected that the reduced predictability due to randomly varied SOAs would only diminish and not eliminate the amplitude of the MMN for both statistical (low vs high) and physical deviants (left vs right). The results show that non-isochronicity reduced stimulus predictability to the point that only physical deviants elicited an MMN, whilst statistical deviants did not. That is, during non-isochronous stimulation the brain responses to high- and low-probability triplet endings were not significantly different, indicating that participants did not track these regularities. The observation that no sMMN was elicited under non-isochronous stimulation reveals the different nature of the neural traces and the functional operations engaged during the elicitation of the sMMN compared to the "classical" MMN observed in traditional oddball paradigms [19, 20, 22, 23, 40]. As suggested previously [25, 26], the sMMN is a neurophysiological marker for deviance detection in regard to the *learned* structural properties of sequential stimuli, namely the probability of a stimulus item given (two) preceding stimulus items. The detection of such local dependencies is not occurring instantaneously, as is the case for the detection of acoustical features in traditional oddball studies, but necessitates longer exposition (i.e. learning) so that individuals can form neural representations of these dependencies.

Contrary to the statistical deviants, the physical deviants elicited a location MMN during non-isochronous stimulation, with a scalp distribution predominantly over medial-frontal sites (cf. [41]). This finding is in agreement with previous studies assessing change detection with irregular temporal structure [19, 20, 22, 23, 40]. While auditory information ascends the cortical hierarchy, physical features are processed at a lower level (e.g. primary auditory cortex) in contrast to the structural features that are processed at a higher level and are likely more prone to temporal manipulations. In addition, the saliency of the physical deviance, namely sound direction, presumably contributes to the elicitation of the location MMN, even though there was a high level of uncertainty about the onset of the stimuli.

An important finding was the interaction between auditory deviants and isochronicity. Error-signals for both statistical and physical deviants were significantly reduced when stimuli presentation was non-isochronous. The comparison of the elicited error-signals during isochronous and non-isochronous stimulation presents great interest from a PC perspective. During non-isochronous stimulation, participants were not able to predict when the next stimulus would be presented. Notably, the induced uncertainty (high entropy) regarding the *timing* of the upcoming events subsequently affected the predictions about the *content* of the events, namely what tones were presented. We suggest that increasing the entropy about the "when" of the events impaired the first-order predictions of the "what" (as indicated by the reduced sMMN), and also decreased the *overall* precision of the model (i.e., impaired the second-order predictions *in general*), leading to an impaired prediction of the "where" (as indicated by the reduced location MMN). The elicitation of a prediction error to deviants requires an a-priori establishment of an accurate prediction about the standards [42]. Under non-isochronous stimulation predictions for standard triplets became imprecise and thus deviant detection mechanisms were impeded. Thus, the significant reduction of the sMMN amplitude is to a large extent driven by the difference in ERPs to standards between the two studies (see S1 Fig), and this indicates that isochronous, or perhaps at least regular, stimulation is likely a prerequisite to form a stimuli memory trace of the statistical regularities. Our findings thus corroborate previous studies showing attenuated prediction error responses when precision diminishes [9, 10, 40, 43, 44].

Although the transitional probabilities between the tones were identical in the two experiments, the current results suggest that participants could not construct a probabilistic model for the statistical deviants under non-isochronous stimulation. According to the 'Bayesian coding hypothesis', the brain represents sensory information probabilistically [45], namely computing probability distributions of the occurring events. Therefore one could expect that participants would construct an almost identical probabilistic model regarding the content under both stimulation modes (isochronous/non-isochronous). The present findings suggest the opposite: the processing of the "what" is tightly interconnected to the processing of the "when". In other words, if we cannot make predictions on the "when" then this impacts (negatively) on our making predictions on the "what". The precision of a distribution decreases with its entropy and under non-isochronous mode the SOA probability distribution became imprecise, and likely influenced the precision of the probability distribution of "what" and "where". Our findings are in line with a previous study showing an interaction of "what" and "when" predictability, giving rise to reduced amplitudes of auditory evoked responses when using random SOAs [46]. We argue that stimulation timing, a seemingly irrelevant dimension for learning performance, influences the learning outcomes, and thus may be regarded as an important factor within fundamental models of perception and learning, such as PC or computational simulations of implicit learning. Music is a domain where temporal and structural expectations are tightly intertwined [5, 47, 48], and the IDyOM model of musical melody processing has integrated the inter-onset-interval as one of the models parameters [49].

The current findings shed light on the interaction of sensory systems with synchronisation and learning. They motivate further research especially considering the fact that the primary auditory difficulties for children with reading or language problems appear to involve rhythmic processing [50]. Human brain capacities in regard to temporal processing are limited and it seems that regularity facilitates encoding and therefore predictions about upcoming events [51, 52]. Temporal regularity is a predominant feature of communication systems, and as suggested by Lumaca et al. [53] it is rooted in the neural constraints for information processing. Additionally, according to dynamic attending theory [54] our attending levels tend to oscillate in synchrony (entrainment) with the periodicities of external events. In light of the PC

framework, a steady beat can be considered as a factor that increases the precision of predictions and thus the attentional gain. In the current study the non-isochronous stimulation probably prevented participants from attending selectively to information which could possibly resolve uncertainty [5].

## Conclusion

We demonstrated that impeded predictability of one stimulus aspect impedes processing of error signals for other stimulus aspects and this, in turn, impedes learning. Even though the temporal aspect, namely the "when" of the stimuli was not part of the learning material, it nevertheless modulated learning of the "what". We suggest that the ensued temporal unpredictability, due to random SOAs, impaired the precision of the predictive model, leading to reduced prediction error signals as reflected in the reduced amplitudes of the elicited sMMN and location MMN. The current results shed new light on the brain's coupled predictive and learning mechanisms. We showed that manipulating stimuli predictability over the time dimension which seems irrelevant for the learning, ultimately impeded the formation of precise expectations and interfered with statistical learning. In a traditional computational model of implicit learning, SOAs are not part of the regularities underlying the simulated learning process. On the other hand, PC provides a larger explanatory power by suggesting that sensory systems by default take into account all attributes of an event. Under non-isochronous stimulation, the task of representing all the aspects of the stimuli becomes computationally more demanding for the predictive brain and thus affects the learning outcomes. Our results contribute to a better understanding of the implicit statistical learning mechanisms by showing that human listeners form predictions based on all aspects of a stimulus, and that varying the predictability of one of these aspects influences the learning of the others.

## Supporting information

**S1 Fig. ERPs of standards.** Mean ERP waves for standards (high-probability) endings under non isochronous and non-isochronous stimulation as captured at electrode Fz.
(TIFF)

## Acknowledgments

The authors would like to thank Vivien Rieder for helping in conducting the experiment. The first author would like to thank Andrea Bischoff, Lucy Werner and Nadia Tsogli for contributing valuable feedback to the manuscript.

## Author Contributions

**Conceptualization:** Stefan Koelsch.

**Data curation:** Vera Tsogli, Sebastian Jentschke, Stefan Koelsch.

**Software:** Vera Tsogli, Sebastian Jentschke.

**Supervision:** Sebastian Jentschke, Stefan Koelsch.

**Writing – original draft:** Vera Tsogli, Stefan Koelsch.

**Writing – review & editing:** Vera Tsogli, Sebastian Jentschke, Stefan Koelsch.

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
