## [Decision Letter · Decision Letter 0]

21 Apr 2021

PONE-D-21-07718

Unpredictability of the “when” impedes learning of the “what” and “where”

PLOS ONE

Dear Dr. Tsogli,

Thank you for submitting your manuscript to PLOS ONE. After careful consideration, we feel that it has merit but does not fully meet PLOS ONE’s publication criteria as it currently stands. Therefore, we invite you to submit a revised version of the manuscript that addresses the points raised during the review process.

We look forward to receiving your revised manuscript.

Kind regards,

Jyrki Ahveninen

Academic Editor

PLOS ONE

Journal Requirements:

2. Please change "female” or "male" to "woman” or "man" as appropriate, when used as a noun (see for instance https://apastyle.apa.org/style-grammar-guidelines/bias-free-language/gender).

Reviewers' comments:

Reviewer's Responses to Questions

**Comments to the Author**

1. Is the manuscript technically sound, and do the data support the conclusions?

Reviewer #1: Yes

Reviewer #2: Yes

2. Has the statistical analysis been performed appropriately and rigorously? 

Reviewer #1: Yes

Reviewer #2: Yes

3. Have the authors made all data underlying the findings in their manuscript fully available?

Reviewer #1: No

Reviewer #2: Yes

4. Is the manuscript presented in an intelligible fashion and written in standard English?

Reviewer #1: Yes

Reviewer #2: Yes

5. Review Comments to the Author

Reviewer #1: This is an interesting paper on the role of temporal predictions on predictions of statistical structure and physical location. The question is an important and difficult one, as indeed, time is often neglected in the statistical learning and predictive coding framework. The protocol is sound and the analyses as well. The paper is overall clear. I have a few suggestions to strengthen the message and clarify the overall structure.

1.

I think that the introduction would gain from a somewhat deeper theoretical perspective. For instance, the authors could cite the work of Dehaene et al 2015 published in Neuron. There the authors suggest five distinct systems capable of representing sequence knowledge at increasing degrees of abstraction. The first level is transition and timing and it serves as a base for the following levels. This is quite different from what the authors claim in the introduction, namely that changes in transition timing should not affect statistical learning.

Dehaene, S., Meyniel, F., Wacongne, C., Wang, L., & Pallier, C. (2015). The neural representation of sequences: from transition probabilities to algebraic patterns and linguistic trees. Neuron, 88(1), 2-19.

2.

On a more “technical” note I think that the last section of the introduction is less clear than the rest of the introduction. Some restructuring may be necessary from line 56. It would also be important to explain why the authors used two types of deviant stimuli (what and where). Did they have different hypotheses concerning the effect of isochrony on stimulus type deviancy ? For instance, a stronger effect on statistical deviancy (cognitive) and smaller on location (lower level perception)? This is what I would expect considering that statistical structure of auditory sequences is temporal, while location is not.

3.

When reading the methods the rationale for using Shepard's tones is not clear. Is this really important also considering that the authors correctly permute the stimuli/"letter" relation across participants.

4.

The result section would be much simpler if the authors run a global model. As it is, it is difficult to grasp the results. Isochronicity should be a factor and deviancy type should be another one. If the model becomes too big, the authors could focus on a literature based topographical region of interest (mean of ROI). Indeed the partitioning into anteroposterior and centrolateral ROIs does not seem to add much to the results(although statistical deviancy seemsto be slightly more frontal).

5.

In the discussion, in reference to what I stated above (point 1), I would tend to complexify a bit the first interpretation of the results, namely "the greater saliency of the physical deviance, in contrast to the statistical deviance, presumably explains the elicitation of the location MMN even when there was a high level of uncertainty about the onset of the stimuli." While this hypothesis is a good one, it is also possible that this difference is due to the nature of the deviancy. A statistical deviancy is temporal, while this is not the case for a location deviancy. This may (also/possibly) explain why the location deviancy is more robust to non-isochronicity.

6.

The informational content hypothesis is interesting although a mixed-model analysis including single trial informational content as a regressor would be more convincing. I do not know if the authors tried this, it should not be too complicated and it would be highly relevant since the discussion brings a lot on the role of informational content that, for now, is only "indirectly" assessed.

I congratulate the authors for their work and I hope that my remarks will allow them to clarify and strengthen their message.

Best regards

Daniele Schön (signed review)

Reviewer #2: In this interesting study, the authors address the role of temporal regularity in the statistical learning of sound patterns using electroencephalography (EEG). Participants were presented with a continuous stream of standard and deviant sound triplets, where sound onsets asynchronies (SOAs) were randomized in the range 0-300 ms. The deviations were either (i) statistical (low-probability), (ii) physical (presented counter-laterally), or (iii) double (low-probability; counter-lateral). Critically, the authors compared their MMN results with a nearly identical experiment where sound triplets were presented with a regular timing (i.e., isochronous SOAs). The comparison showed a relative attenuation in the physical ("where") and statistical ("what") MMN responses for temporally irregular stimuli. This result was interpreted with the precision-weighting hypothesis of the predictive coding (PC) framework: statistical and physical predictions are less reliable in a temporally irregular context, resulting in down-weighted neural prediction errors. The conclusion is that temporal regularity is a fundamental prerequisite of statistical learning, that so far has been neglected in computational models. I liked the manuscript: it is well written and discussed. Below you can find a few suggestions and some concerns on the methodology/results that the authors could address.

Methodology

- It could be useful to add tables to highlight the main differences between the two studies: e.g., one table with demographic information (optional) and another with key aspects of stimulation/analyses. The reader will see immediately that most parameters match, except for the factor of interest (temporal regularity: irregular vs isochronous) and few other parameters (e.g., sound duration: 150 vs. 220; sound timbre: 1 vs. 6 percussions; software for statistical analysis: JASP vs. SPSS; etc.). My opinion is that these tables will give the article more clarity: it is, after all, a comparison study. Further, it will be easier (and faster) for the readers to decide whether the few differences might or not be factors that affect the comparison.

- The authors could also present individual waveform data and grand averages with 95% confidence intervals (CI). This would require some additional work. However, it would be useful for the readers to assess how many participants show the results you described in the manuscript.

- I understand that this study must be consistent, in methodology and analyses, with the previous experiment [25]. I find, however, that EEG analyses limited to a few clusters of electrodes and to one specific time window, are a bit outdated… I wonder why it was not performed an ANOVA for all time points and electrodes so that effects could be revealed in time and space without any biases. This can be achieved e.g. in EEGlab using the Fmax permutation method (Groppe et al., 2011). No fear, I am not suggesting to do new analyses. However, I feel that this is a methodological weakness that could have been easily addressed (perhaps in future works…).

- Which criteria did you use to define the second time window for statistical MMN? One method is to centre a time window on the peak of the grand-average difference wave, but you might have used another method. Please specify.

Results

- I might have overlooked this point: Why are the results for the double deviants missing in both studies? What are the authors’ expectations in relation to the interaction between the two deviant types?

- Temporal irregularity is shown to reduce MMN responses, and this result was interpreted as suppression of prediction errors from precision-weighting mechanisms. One alternative hypothesis is that, in the present condition, there might be more contamination from neural responses of the previous sounds. This would occur if the SOA between the penultimate and the last sound of the triplets is small (e.g., 50 ms). In the manuscript, it is mentioned briefly that a form of selection was performed on standard and deviant epochs so that “only [those] with at least 200 ms from the adjacent trigger were included in the analysis”. Is this the way to address the mentioned issue? If that is the case, I would suggest to say it more explicitly.

6. PLOS authors have the option to publish the peer review history of their article (what does this mean?). If published, this will include your full peer review and any attached files.

Reviewer #1: **Yes: **Daniele Schön

Reviewer #2: **Yes: **Massimo Lumaca

---

## [Author Response · Author response to Decision Letter 0]

16 Jun 2021

Response to the editor’s and reviewers’ comments 

Journal Requirements:

Comment 1. Please ensure that your manuscript meets PLOS ONE's style requirements, including those for file naming. The PLOS ONE style templates can be found at 

Response: The manuscript has been revised according to the journal style requirements (please see the revised manuscript).

Comment 2. Please change "female” or "male" to "woman” or "man" as appropriate, when used as a noun (see for instance https://apastyle.apa.org/style-grammar-guidelines/bias-free-language/gender).

Response: This manuscript has been revised according to the journal style requirements (please see the revised manuscript).

Comment 3. We note that you have indicated that data from this study are available upon request. PLOS only allows data to be available upon request if there are legal or ethical restrictions on sharing data publicly. For more information on unacceptable data access restrictions, please see http://journals.plos.org/plosone/s/data-availability#loc-unacceptable-data-access-restrictions.

Response: Your request was considered but we are not able to satisfy it because we are bound by the consent form that participants signed before the experiment, which states that “The information that is recorded about you will only be used as described in the purpose of the study.”. Participants have not given their consent on any kind of public sharing of their data. We acknowledge that this is a weakness of our study and we will try to amend it in our future projects. The consent form has been approved by the Regional Committee for Medical and Health Research Ethics which has reviewed and approved the Research Project, 2018/2409 (https://helseforskning.etikkom.no/forside?_ikbLanguageCode=n).

Response to Reviewer #1 comments:

Comment 1: I think that the introduction would gain from a somewhat deeper theoretical perspective. For instance, the authors could cite the work of Dehaene et al 2015 published in Neuron. There the authors suggest five distinct systems capable of representing sequence knowledge at increasing degrees of abstraction. The first level is transition and timing and it serves as a base for the following levels. This is quite different from what the authors claim in the introduction, namely that changes in transition timing should not affect statistical learning.

Response: The reviewer raised a very interesting point by citing the work of Dehaene et al., 2005. We found this research work very relevant to our study and tried to take it into account in our revised manuscript. Thus, we have modified our Introduction (lines 91 to 104). The corresponding excerpt from the revised manuscript follows:

"By doing so, our study also aimed at adding evidence to the existing theoretical frameworks describing the cognitive processes during sequence learning. For instance, computational models of implicit learning neglect stimulus timing because sequential models (i.e., n-grams or Markov models) focus on the ordering of the events [16,17]. On the other hand, PC suggests that prediction-driven learning relies on the estimation of probabilities [30], but it remains unclear whether modelling of one stimulus dimension (e.g., pitch) interacts with another stimulus dimension (e.g., timing). Deheane et al. [31] have proposed a theoretical framework for sequence learning with increasing levels of abstraction. Interestingly, transitions and timing knowledge are placed at the first level, suggesting the tight interconnection between the learning of the “what” and the “when”. This supports our hypothesis that unpredictable (non-isochronous) stimulus onsets would impede accurate predictions, and thus impair learning as reflected in the amplitudes of the statistical and the location MMN."

Comment 2: On a more “technical” note I think that the last section of the introduction is less clear than the rest of the introduction. Some restructuring may be necessary from line 56. It would also be important to explain why the authors used two types of deviant stimuli (what and where). Did they have different hypotheses concerning the effect of isochrony on stimulus type deviancy ? For instance, a stronger effect on statistical deviancy (cognitive) and smaller on location (lower level perception)? This is what I would expect considering that statistical structure of auditory sequences is temporal, while location is not.

Response: We would like to thank the reviewer for the useful remarks. We agree that the last paragraph of the Introduction was not as clear as the rest and thus we have changed it accordingly from line 56 up to line 91 (excerpt follows). In the additional lines we have addressed all the questions raised by the reviewer. The corresponding excerpt from the revised manuscript follows: 

"To investigate how statistical learning and auditory deviance detection are affected by temporal predictability we employed a variation of an experimental paradigm used in a previous study [25]. We constructed a continuous auditory stream of sound triplets with deviants that were either (a) statistical, in terms of transitional probability, (b) physical, due to location change (“standards" were presented from one direction, whilst “deviants" were presented from the other direction) or (c) double deviants, i.e. a combination of the two (see Fig 1). Statistical and physical deviants tapped different stimulus dimensions. Specifically, statistical deviants regarded the stimuli content, or the “what", whilst the physical deviants regarded the stimuli location, or the “where". Contrary to our previous study [25], where a constant SOA was used, in the current study we used random SOAs as a means to manipulate the predictability of stimulus-onset. In that way, we manipulated a third stimulus dimension which was the time, or the “when".

 Our aim was to examine how effects of predictability are reflected in two ERP components, the statistical MMN (sMMN) [25, 26] and the location MMN. The sMMN was used as a neurophysiological marker of statistical learning and the location MMN as a marker of auditory deviance detection. We tested whether statistical or physical deviants would elicit a sMMN or a location MMN respectively, even when SOAs are random. A secondary scope of our study was to compare the results from the current study (using random SOAs) and our previous experiment using isochronous SOAs [25]. We expected that both sMMN and location MMN would be affected by the predictability of the SOA, and therefore we tested for an interaction between the ERP-effects elicited by deviant events (deviants compared to standards) and timing 

(isochronous vs. non-isochronous). Within the PC framework, unpredictability (computed as Shannon entropy) expresses the level of uncertainty about upcoming events [27]. For instance, low entropy is indicative of highly probable upcoming events [6]. Compared to our previous study, in the current paradigm the uncertainty or the entropy of the “when” changed from minimal (isochronous events with no uncertainty) to maximal (random SOAs with high uncertainty [28, 29]), whereas the entropy in terms of the “what” (what tones are presented) and “where” (sound direction) remained the same. Therefore, we expected that manipulating the entropy of the “when” (by switching from isochrony with zero entropy in the temporal structure, to randomly varied SOAs with maximal entropy) would impact the predictive processes 

of “what” and “where”. In PC terms, this manipulation (switching to random SOAs) reduced precision of the predictive model, which we expected to be reflected in a reduction of the amplitudes of error signals (the sMMN and the location MMN)." 

Comment 3. When reading the methods the rationale for using Shepard's tones is not clear. Is this really important also considering that the authors correctly permute the stimuli/"letter" relation across participants.

Response: We are grateful for the reviewer’s remark. The decision for using Shepard's tones was taken after conducting pilot testing with pure tones, during which it came to our attention that entrainment to the triplet structure was inhibited by auditory grouping due to pitch perception. Thus, the choice for Shepard’s tones was made. In the revised manuscript we reformulated the part in the Methods (lines 123 to 125) where we explain how we created the sounds. The corresponding excerpt from the revised manuscript follows:

"The use of pure tones was rejected during pilot testing because emergent Gestalt formation (i.e., ascending or descending triads) were confounding learning of the triplet structure [33, 34]."

Comment 4. The result section would be much simpler if the authors run a global model. As it is, it is difficult to grasp the results. Isochronicity should be a factor and deviancy type should be another one. If the model becomes too big, the authors could focus on a literature based topographical region of interest (mean of ROI). Indeed the partitioning into anteroposterior and centrolateral ROIs does not seem to add much to the results(although statistical deviancy seems to be slightly more frontal).

Response: Thank you for the remark. We would like to note that the results section presents first the results from the new experiment (with random SOAs) and then a comparison between the new and the previous experiment (with a constant SOA). Further down, in sections “Interaction of statistical deviance and isochronicity” and “Interaction of physical deviance and isochronicity” we describe the two global ANOVAs that we conducted to compare the ERPs from the two experiments. The reason for not running a single global ANOVA (where deviance type would be one of the factors) is that it was not in our hypotheses to compare the two deviances.

Nonetheless, the reviewer's remark led us to modify the Methods section (lines 338 to 341 and lines 381 to 382). The corresponding excerpts from the revised manuscript follow:

"Repeated measures analyses of variance (ANOVAs) were carried out to assess the ERPs of the current experiment with random SOAs and then compare the results between the current experiment and the previous one where a constant SOA was used."

"First, we will present the ERPs of statistical and physical deviants under the non-isochronous stimulation of the current experiment. "

Comment 5. In the discussion, in reference to what I stated above (point 1), I would tend to complexify a bit the first interpretation of the results, namely "the greater saliency of the physical deviance, in contrast to the statistical deviance, presumably explains the elicitation of the location MMN even when there was a high level of uncertainty about the onset of the stimuli." While this hypothesis is a good one, it is also possible that this difference is due to the nature of the deviancy. A statistical deviancy is temporal, while this is not the case for a location deviancy. This may (also/possibly) explain why the location deviancy is more robust to non-isochronicity.

Response: The authors agree in part with the reviewer’s remark. We would like to note that neither the statistical nor the physical deviants are temporal. The temporal manipulation was identical for both deviant types but the effect was different. Our findings point to the direction that physical deviants are more robust to temporal manipulations compared to statistical deviants. We have tried to underline this difference and thus we have changed the corresponding part in the Discussion (lines 522 to 529). The corresponding excerpt from the revised manuscript follows: 

"This finding is in agreement with previous studies assessing change detection with irregular temporal structure [19,20,22,23,40]. While auditory information ascends the cortical hierarchy, physical features are processed at a lower level (e.g. primary auditory cortex) in contrast to the structural features that are processed at a higher level and are likely more prone to temporal manipulations. In addition, the saliency of the physical deviance, namely sound direction, presumably contributes to the elicitation of the location MMN, even though there was a high level of uncertainty about the onset of the stimuli."

Comment 6. The informational content hypothesis is interesting although a mixed-model analysis including single trial informational content as a regressor would be more convincing. I do not know if the authors tried this, it should not be too complicated and it would be highly relevant since the discussion brings a lot on the role of informational content that, for now, is only "indirectly" assessed.

Response: We agree with the point that the reviewer raised. Referring to the IC values for statistical and physical deviants was done only for discussion purposes. But, in the revised manuscript we have deleted the corresponding sentences from the Discussion (lines 495 to 502 on the previous version of the manuscript). We are thankful for the reviewer's remark and we agree that including single-trial analysis adds a different perspective in a study and we intend to take it into account in our future studies. 

Response to Reviewer #2 comments:

Comment 1. It could be useful to add tables to highlight the main differences between the two studies: e.g., one table with demographic information (optional) and another with key aspects of stimulation/analyses. The reader will see immediately that most parameters match, except for the factor of interest (temporal regularity: irregular vs isochronous) and few other parameters (e.g., sound duration: 150 vs. 220; sound timbre: 1 vs. 6 percussions; software for statistical analysis: JASP vs. SPSS; etc.). My opinion is that these tables will give the article more clarity: it is, after all, a comparison study. Further, it will be easier (and faster) for the readers to decide whether the few differences might or not be factors that affect the comparison.

Response: We would like to thank the reviewer for this useful remark! Our revised manuscript reflects this suggestion. We have added three new tables where a comparative view between the two studies is presented in regard to: a) demographic information, b) experiment design details and c) methodological details.

Comment 2. The authors could also present individual waveform data and grand averages with 95% confidence intervals (CI). This would require some additional work. However, it would be useful for the readers to assess how many participants show the results you described in the manuscript.

Response: Yes, we agree, this is a good point! In our revised manuscript we have adjusted Fig 2 by presenting the ERP waveforms with standard error of the means.

Comment 3. I understand that this study must be consistent, in methodology and analyses, with the previous experiment [25]. I find, however, that EEG analyses limited to a few clusters of electrodes and to one specific time window, are a bit outdated… I wonder why it was not performed an ANOVA for all time points and electrodes so that effects could be revealed in time and space without any biases. This can be achieved e.g. in EEGlab using the Fmax permutation method (Groppe et al., 2011). No fear, I am not suggesting to do new analyses. However, I feel that this is a methodological weakness that could have been easily addressed (perhaps in future works…).

Response: We would like to thank the reviewer for his useful feedback which we intend to take it into consideration in our future studies. In addition to what is described in our manuscript, we would like to note that our analysis script indicated the time periods where the difference between the conditions is significant and subsequently after visual inspection we selected a time window, within the time periods indicated by the script, to perform the ANOVA.

Comment 4. Which criteria did you use to define the second time window for statistical MMN? One method is to centre a time window on the peak of the grand-average difference wave, but you might have used another method. Please specify.

Response: We would like to thank the reviewer for pointing this out. The second time-window for the statistical MMN (150 to 200 ms) was selected because during that period the difference between the waveforms appeared to be largest. We decided to go for a tight time window so as to reveal any existing effect. In the revised manuscript we took into account the reviewer’s comment and we reformulated the corresponding part in the Results (lines 406 to 410). The corresponding excerpt from the revised manuscript follows:

"To ensure that the present finding of no effect of triplet ending is not simply due to the choice of the time window from 180 to 260 ms (which was based on our previous study; [25]), the analysis was repeated over a tighter time window from 150 to 200 ms, during which the difference between the waveforms appeared to be largest."

Comment 5. I might have overlooked this point: Why are the results for the double deviants missing in both studies? What are the authors’ expectations in relation to the interaction between the two deviant types?

Response: In our previous study with isochronous stimulation [25], we described the effects of the double deviants in the section “Interaction of probability and change location”. Our finding was that “At the presence of physical MMN the statistical MMN diminishes” (Tsogli et al., 2019). In the current study, we did not make any hypotheses regarding the interaction of the two deviant types. 

Comment 6. Temporal irregularity is shown to reduce MMN responses, and this result was interpreted as suppression of prediction errors from precision-weighting mechanisms. One alternative hypothesis is that, in the present condition, there might be more contamination from neural responses of the previous sounds. This would occur if the SOA between the penultimate and the last sound of the triplets is small (e.g., 50 ms). In the manuscript, it is mentioned briefly that a form of selection was performed on standard and deviant epochs so that “only [those] with at least 200 ms from the adjacent trigger were included in the analysis”. Is this the way to address the mentioned issue? If that is the case, I would suggest to say it more explicitly.

Response: We would like to thank the reviewer for this useful comment. Indeed, the SOA in the present study can be as small as 50 ms and this can possibly contaminate the ERPs. In the revised manuscript we took into account the reviewer’s comment and we reformulated the part in the Methods (lines 317 to 320) where we explain the EEG preprocessing routine. The corresponding excerpt from the revised manuscript follows:

"In that way we ensured that only tones with at least 50 ms apart from the adjacent one (either before or after) would be included in the analysis and thus reduce any possible contamination of ERP responses from adjacent sounds occurring too early."

---

## [Decision Letter · Decision Letter 1]

27 Oct 2021

PONE-D-21-07718R1Unpredictability of the “when” impedes learning of the “what” and “where”PLOS ONE

Dear Dr. Tsogli,

Thank you for submitting your manuscript to PLOS ONE. After careful consideration, we feel that it has merit but does not fully meet PLOS ONE’s publication criteria as it currently stands. Therefore, we invite you to submit a revised version of the manuscript that addresses the points raised during the review process. EDITORIAL COMMENT: The response to the data sharing requirement is incomplete. There are ways to make measured data unidentifiable: please consult your ethical committee. The analysis scripts and code could also be shared.

We look forward to receiving your revised manuscript.

Kind regards,

Jyrki Ahveninen

Academic Editor

PLOS ONE

Journal Requirements:

Reviewers' comments:

Reviewer's Responses to Questions

**Comments to the Author**

1. If the authors have adequately addressed your comments raised in a previous round of review and you feel that this manuscript is now acceptable for publication, you may indicate that here to bypass the “Comments to the Author” section, enter your conflict of interest statement in the “Confidential to Editor” section, and submit your "Accept" recommendation.

Reviewer #2: All comments have been addressed

2. Is the manuscript technically sound, and do the data support the conclusions?

Reviewer #2: Yes

3. Has the statistical analysis been performed appropriately and rigorously? 

Reviewer #2: Yes

4. Have the authors made all data underlying the findings in their manuscript fully available?

Reviewer #2: No

5. Is the manuscript presented in an intelligible fashion and written in standard English?

Reviewer #2: Yes

6. Review Comments to the Author

Reviewer #2: The authors addressed all my comments. I have only few minor suggestions for tables and figures.

Tables

- All tables. I'd remove "previous study" and use the correct reference ("Tsogli et al. 2019").

- Table 2. Terminology. It is subtle the difference in meaning between "interstimulus interval" and "SOA". I understand that "interstimulus interval" is the silent interval between the offset of one tone and the onset of the next tone within the same triplet (I could not find this definition in the main text). "SOA" is instead the silent interval between tone onsets within the same triple. However, several studies define interstimulus interval as the time interval between the offset of the last tone in a triplet and the onset of the first tone in the next triplet. Why not simply add the SOA range to avoid confusion? Then, I would include the 'real' interstimulus interval (as defined in the previous point), which is currently missing.

- Table 4. Please, correct "current study" with "Tsogli et al. 2019" (under box: Isochronous stimulation).

Figures

- Fig. 1. The last example triplet of the train is a deviant (CDE).

- Fig. 2. ERP results pertain to two different experiments/studies and this should be made clearer. It could be useful for readers who skim through the figures, deciding whether or not to read the main text. At first glance, it appears that both conditions were performed in this study (ISO and non-ISO).

- Fig. 2. The shaded area on the ERPs represents the SEM. Please, include this information in the legend.

Main text:

line 342. “EPRs”

7. PLOS authors have the option to publish the peer review history of their article (what does this mean?). If published, this will include your full peer review and any attached files.

Reviewer #2: **Yes: **Massimo Lumaca

---

## [Author Response · Author response to Decision Letter 1]

26 Dec 2021

Response to the editor’s and reviewers’ comments

Journal Requirements:

Comment 1. The response to the data sharing requirement is incomplete. There are ways to make measured data unidentifiable: please consult your ethical committee. The analysis scripts and code could also be shared.

Response:

We agree that in our submitted version we were not able to satisfy your request to make the data publicly available. In the new version of our manuscript we provide a link to our data which we made them unidentifiable.

Response to Reviewer comments:

Comment 1: The authors addressed all my comments. I have only few minor suggestions for tables and figures.

Response: 

We are very grateful to the useful suggestions made by the reviewer. In our revised paper we have made all the necessary changes to address all the reviewer’s remarks regarding the tables, the figures and the text.

Specifically, in the tables:

- we have replaced references to the “previous study” with (“Tsogli et al, 2019”).

- we have added the SOA range, the ‘real’ interstimulus interval. Additionally on the Table legend we have added definitions for the SOA and interstimulus interval to avoid confusion:

Interstimulus interval denotes the silent interval between the offset of one tone and the onset of the next one. SOAs denote the interval between the onset of one tone and the onset of the next one

- on Table 4, we have replaced "current study" with "Tsogli et al. 2019" (under box: Isochronous stimulation).

In the figures:

- we have corrected the last example triplet of the train from CDE to CDF.

- we have changed the legend title of Figure 2 so that it is clear to the reader that the depicted results pertain to two different experiments/studies. The new legend title follows:

ERPs of statistical and physical deviants as recorded in the current study with non-isochronous sound presentation, and in the previous study [25] with isochronous sound presentation.

- we have modified the legend of Figure 2 to denote that the shaded area on the ERPs represent SEM. The new legend follows:

The shaded area on the ERPs represents the SEM.

In the text:

- We have modified line 342 of the manuscript where we mistakenly wrote “EPRs” instead of “ERPs”.

---

## [Editor Report · Decision Letter 2]

19 Jan 2022

Unpredictability of the ''when'' influences prediction error processing of the ''what'' and ''where''

PONE-D-21-07718R2

Dear Dr. Tsogli,

We’re pleased to inform you that your manuscript has been judged scientifically suitable for publication and will be formally accepted for publication once it meets all outstanding technical requirements.

Kind regards,

Jyrki Ahveninen

Academic Editor

PLOS ONE
---

## [Editor Report · Acceptance letter]

26 Jan 2022

PONE-D-21-07718R2 

Unpredictability of the “when” influences prediction error processing of the “what” and “where”. 

Dear Dr. Tsogli:

I'm pleased to inform you that your manuscript has been deemed suitable for publication in PLOS ONE. Congratulations! Your manuscript is now with our production department. 

Kind regards, 

on behalf of

Dr. Jyrki Ahveninen 

Academic Editor

PLOS ONE